# Radiation as a Tool against Neurodegeneration—A Potential Treatment for Amyloidosis in the Central Nervous System

**DOI:** 10.3390/ijms232012265

**Published:** 2022-10-14

**Authors:** Carina Marques Coelho, Lia Pereira, Pamela Teubig, Pedro Santos, Filipa Mendes, Sílvia Viñals, Daniel Galaviz, Federico Herrera

**Affiliations:** 1Laboratório de Instrumentação e Física Experimental de Partículas (LIP), Av. Prof. Gama Pinto 2, 1649-003 Lisboa, Portugal; 2BioISI—Instituto de Biosistemas e Ciências Integrativas, Faculdade de Ciências, Universidade de Lisboa, 1749-016 Lisboa, Portugal; 3Centro de Ciências e Tecnologias Nucleares (C2TN), Instituto Superior Técnico, Universidade de Lisboa, E.N. 10 ao km 139.7, 2695-066 Bobadela LRS, Portugal; 4Departamento de Engenharia e Ciências Nucleares (DECN), Instituto Superior Técnico, Universidade de Lisboa, Estrada Nacional 10, 2695-066 Bobadela LRS, Portugal; 5Centro de Micro-Análisis de Materiales (CMAM), Universidad Autónoma de Madrid (UAM), Campus de Cantoblanco, C/Faraday 3, 28049 Madrid, Spain; 6Departamento de Física, Faculdade de Ciências, Universidade de Lisboa, Campo Grande 016, 1749-016 Lisboa, Portugal; 7Departamento de Química e Bioquímica, Faculdade de Ciências, Universidade de Lisboa, Campo Grande 016, 1749-016 Lisboa, Portugal

**Keywords:** amyloid, neurodegeneration, radiation therapy

## Abstract

Radiotherapy (RT) is a relatively safe and established treatment for cancer, where the goal is to kill tumoral cells with the lowest toxicity to healthy tissues. Using it for disorders involving cell loss is counterintuitive. However, ionizing radiation has a hormetic nature: it can have deleterious or beneficial effects depending on how it is applied. Current evidence indicates that radiation could be a promising treatment for neurodegenerative disorders involving protein misfolding and amyloidogenesis, such as Alzheimer’s or Parkinson’s diseases. Low-dose RT can trigger antioxidant, anti-inflammatory and tissue regeneration responses. RT has been used to treat peripheral amyloidosis, which is very similar to other neurodegenerative disorders from a molecular perspective. Ionizing radiation prevents amyloid formation and other hallmarks in cell cultures, animal models and pilot clinical trials. Although some hypotheses have been formulated, the mechanism of action of RT on systemic amyloid deposits is still unclear, and uncertainty remains regarding its impact in the central nervous system. However, new RT modalities such as low-dose RT, FLASH, proton therapy or nanoparticle-enhanced RT could increase biological effects while reducing toxicity. Current evidence indicates that the potential of RT to treat neurodegeneration should be further explored.

## 1. Introduction

Neurodegenerative disorders, such as Alzheimer’s Disease (AD), Parkinson’s Disease (PD), Huntington’s Disease (HD), Amyotrophic Lateral Sclerosis or Familial Prion Diseases, are a large and heterogeneous group of age-related disorders involving the progressive loss of specific neuronal populations. The type of neurons and the nervous structure that degenerates first determines the clinical symptoms for each disease, which ranges from memory loss to motor impairment. The mechanisms underlying neuronal loss are still unclear, even in the inheritable, monogenetic versions of these disorders and, consequently, the treatments available are purely symptomatic. However, there are several histopathological hallmarks common to most of these disorders that have been extensively studied, characterized and targeted in experimental and clinical settings.

Neurodegeneration is often caused by forms of programmed cell death, such as apoptosis, necroptosis/oxytosis/ferroptosis or autophagy [1,2]. These forms of cell death are almost invariably associated with oxidative stress, neuroinflammation and alterations in the homeostasis of proteins. Oxidative stress is usually related with mitochondrial dysfunction and an imbalance in the levels of metals, which catalyze the production of free radicals from oxygen and nitrogen. Both mitochondrial dysfunction and metal-catalyzed oxidative stress contribute to neurodegeneration by oxidizing lipids, proteins, and DNA [3,4,5]. Neuroinflammation contributes to the development and exacerbation of AD [3,6,7], PD [8] and other neurodegenerative disorders [9]. It is a response to tissue damage that involves the activation of microglia and astroglia, two defensive/supportive cells necessary for normal neuronal function [3]. Reactive glia grows, proliferates, and cleans up the tissue by removing the fragments of death neurons from the extracellular space, among other functions. Reactive glia can also form a scar in the damaged tissue in extreme cases where the blood–brain barrier is disrupted [10]. Neuroinflammation is essentially a protective mechanism, but it can be noxious if it becomes chronic. Interestingly, a recently found correlation between dementia and previous infections, supports a role for chronic inflammation in neurodegeneration [11]. Aggregation of abnormal proteins is another important pathophysiological hallmark of neurodegenerative disorders. Disruption of protein homeostasis leads to the aberrant accumulation of misfolded proteins, which cannot be degraded, and therefore interfere with normal cell functioning. These protein deposits are frequently called amyloids and can also be found in a wide number of disorders not associated with the central nervous system, such as amyloidosis, diabetes or cancer [12]. Amyloidosis is a heterogeneous group of rare disorders characterized by the deposition of abnormal proteins in cells and tissues throughout the body, which can lead to organ dysfunction and failure [13,14,15].

## 2. Amyloidosis in Neurodegenerative Disorders

The composition and location of toxic amyloids is characteristic of each particular neurodegenerative disease. For example, AD brains show aggregates of Amyloid Precursor Protein (APP) and Tau protein in the hippocampus, while α-synuclein aggregates are characteristic of the substantia nigra in PD brains, and huntingtin forms aggregates in the striatum of HD brains. The structure of aggregation-prone proteins is often rich in sticky β-sheets or intrinsically disordered domains (i.e., without a defined structure in their native state), which have a higher tendency to misfold, self-associate and avoid degradation by the ubiquitin-proteosome or lysosome-autophagy systems [16,17].

Amyloids play a relevant role in the pathogenesis of neurodegeneration, but there is a lack of consensus regarding the exact mechanism [12,17]. Large amyloid fibrils were originally thought to be toxic, but current views point to soluble oligomers as the most toxic amyloid species [17,18,19]. Accumulation of toxic oligomeric species into larger aggregates could be even neuroprotective [20,21]. However, there are questions that remain to be elucidated: while targeting amyloid formation in animal models prevents histopathological features and ameliorates the symptoms [8,16,22], clinical trials have invariably failed to modify the course of these disorders [23,24,25,26]. Anti-amyloidogenic drugs do not induce cognitive improvement in AD and, in some cases, they even aggravated the disease. Interestingly, the same happened with antioxidants, which are widely successful in preventing neurodegeneration and in in vitro cell cultures and in vivo animal models but failed repeatedly in clinical trials [3,5]. While oxidative stress indicators are almost invariably found in the brains of AD, PD and other neurodegenerative disorders, and oxidants often mimic these disorders in vitro and in vivo, low to moderate levels of oxidative stress are essential for cell survival and the normal function of the central nervous system [27]. Moreover, an imbalance of redox mechanisms towards the intracellular accumulation of reductants, such as glutathione or NAPDH, can also be a source of cellular stress and toxicity, known as reductive stress [28]. Reductive stress is found in early stages of development in AD mice, and it diminishes neurogenesis [29] and enhances protein aggregation in cellular models of neurodegeneration [30]. In summary, amyloidogenesis and REDOX status are accepted to be central events to neurodegenerative disorders, but whether they are a cause or a consequence, a trigger or a mere symptom, is still unclear. The discordance between the results obtained with anti-amyloidogenic and antioxidant therapies in animal models and clinical trials is currently a matter of intense discussion in the field.

## 3. Hormesis of Ionizing Radiation

Ionizing radiation has extensively been used for cancer treatment for more than a century, since X-rays and radioactivity were discovered in 1895 and 1896, respectively [31,32]. Ionizing radiation can damage cell components by direct bond breakage or by the production of reactive oxygen species due to both water radiolysis and mitochondria stimulation [33]. In the field of cancer, radiotherapy (RT) research has moved in the direction of optimizing the dose delivered to the tumor cells while sparing healthy tissues. For example, current proton therapy approaches make use of the dosimetric advantages of protons, which allow the deposition of higher energy on the target area [34]. Protons of a certain defined energy slow down as they penetrate the tissue and stop when their entire energy is deposited. Dose deposition produces a very characteristic depth–dose curve, with the point of highest dose being called Bragg Peak. The depth where the peak occurs corresponds to the proton range and is a function of their initial energy. It is possible to choose the initial proton energy based on the depth of the volume to be irradiate, causing minimal damage to healthy tissue and allowing a lower integral dose in the entire body [35,36,37]. In contrast, conventional RT based on electron or photon beams deposits a large fraction of energy in the tissues they cross to enter the tumor, causing undesired toxicity in otherwise healthy cells. Electron/photon-based RT schemes are currently being revised to avoid these issues, and new strategies such as FLASH promise higher specificity and efficiency [37].

However, life on earth has evolved under exposure to natural environmental radiation, and is thus adapted to non-toxic or sub-acute radiation levels. From a therapeutic point of view, ionizing radiation is currently considered hormetic, i.e., have either beneficial or deleterious effects depending on the conditions, especially the dose administered [38,39,40,41]. The nature and degree of the biological effects caused by RT depend on the dose, dose-rate, fractionation dose scheme (i.e., the division of the total dose of radiation to administer into multiple smaller doses, called fractions), and treated volume [42,43,44,45,46]. Unlike conventional RT, low-dose irradiation increases antioxidant levels [47,48,49] and activates DNA repair mechanisms [38], possibly leading to a higher resistance to DNA damage in subsequent irradiation sessions [49]. Low-dose RT induces M2 polarization in macrophages [44] causing decreases both in the adhesion of peripheral blood mononuclear cells and granulocytes [50,51], and the expression/activity of inducible nitric oxide synthase leading to less nitric oxide and reactive oxygen species (ROS) [50,52], among other anti-inflammatory effects [53]. Other potentially beneficial effects of low dose irradiation are induction of cell proliferation and tissue repair, slowing ageing or interfering with cancer development and progression [44,54,55]. Low-dose RT has successfully been applied to treat benign inflammatory diseases [44,54,55] with lower toxic side-effects [50,55]. This wide array of non-toxic biological effects could open new venues for the use of RT in therapeutics beyond cancer. However, low-dose RT regimens are not well established and validated, although some protocols recommend doses between 0.3 and 1.5 Gy over 4–5 fractions [53].

## 4. The Potential of Radiation to Treat Neurodegenerative Disorders

Laser therapy has been used for decades to treat amyloidosis in the skin [56] and the respiratory tract [57]. RT has also been introduced in 1967 as a possible treatment for laryngeal amyloidosis [58,59], and fifteen years later, eyelid amyloidosis was treated with superficial RT [60]. In 1998, localized tracheobronchial amyloidosis was treated with RT in a 67-year-old man [13], and other successful cases were sequentially reported [61,62,63]. RT has also been applied for the treatment of extra-cranial amyloidosis conditions, such as orbital [14], periorbital [15], nasopharyngeal [64], laryngeal [59], lung parenchymal [65] and urinary bladder’s [66]. In general, the treatment doses are delivered in fractions of approximately 2 Gy, with the total dose ranging from 24 to 70 Gy. The use of fractionated dose schemes in RT is common to allow healthy cells to regenerate (Table 1).

Despite the success in treating extra-cranial amyloidosis, RT has not been seriously considered for neurodegenerative disorders until recent years. Neurodegenerative disorders are caused by neuronal loss and, as mentioned above, oxidative stress plays a key role in ageing and neurodegeneration. Thus, the use of RT seemed counterintuitive in this context: RT kills cells and produces oxidative stress, potentially worsening neural deterioration. While this could be the case for high-dose RT used for cancer therapy and peripheral amyloidosis, current evidence indicates that other RT schemes could be beneficial for neurodegenerative disorders [67]. In mice, total-body low-dose radiation (0.1 Gy) was reported to not induce AD or memory impairment [68] and high- and low-dose RT produced two completely different gene expression patterns, with low-dose radiation downregulating genes, which were associated with ageing and AD [69]. Moreover, low-dose radiation stimulates neural stem cell proliferation and differentiation into neurons in the mouse hippocampus [70].

The mechanism of action of RT in decreasing amyloidosis remains unknown. Bistolfi [71] hypothesized that ionizing radiation can directly disrupt the aggregates by breaking the intramolecular H-bonds, which are essential for the β-sheet conformation typical of amyloids. Indirectly, ionizing radiation induces autophagy [72,73], a process by which cells can eliminate toxic proteins and damaged organelles [74]. Irradiation also induces intermolecular or intramolecular crosslinking and, therefore, could promote molecule aggregation [71]. Since the role of protein aggregation in neurodegenerative disorders is still under discussion, it is difficult to predict from a theoretical point of view whether RT could be deleterious or beneficial to neurodegenerative disorders. As shown below, a few in vitro and in vivo studies indicated the potential benefits of ionizing radiation in AD, but there is barely any information about the effects of RT on other neurodegenerative disorders.

### 4.1. In Vitro Studies

SH-SY5Y cultured neurons irradiated with 1 Gy before incubation with the 1-42 fragment of Amyloid beta (Aβ_1-42_) for 24 or 48 h showed an improvement in cell viability compared with non-irradiated cells [75]. Low-dose irradiation inhibited the production of neuroinflammatory cytokines induced by Aβ overexpression in BV-2 microglial cells [75]. In a second study, the same group confirmed the decrease in pro-inflammatory cytokine markers and observed an increase in anti-inflammatory markers after irradiating BV-2 microglial cells with a total dose of 0.5, 1, 2, 3.8 or 5.25 Gy [76]. Aβ-expressing PC12 cells irradiated with a LED system (λ = 640 ± 15 nm) also showed a notable decrease both in amyloid load and apoptosis 24 h after irradiation, depending on LED intensity [77]. In principle, light has only a thermal effect and does not ionize molecules, but it is a promising indicator of the potential influence that other types of radiation could have on protein aggregation and neuronal death.

The perturbation of iron homeostasis has been correlated with several neurological diseases, including AD an PD [78,79]. These abnormal quantities of iron in the brain are related to magnetite (Fe_3_O_4_), an iron oxide metal naturally present in the human brain [80]. Biogenic magnetite nanoparticles were first detected in the human brain over 20 years ago [81], and they were found to be associated with neurodegenerative diseases such as AD, PD and HD [82,83,84,85,86]. Magnetite binds to amyloid deposits, forming a complex that causes a lethal deterioration in neuron function [82,83]. The REDOX-active form of magnetite can produce reactive oxygen species and oxidative stress [80,83,87,88]. When mid/high-Z nanoparticles are irradiated with a high energy ion or proton beam, fluorescent X-rays and low-energy electrons are emitted, a phenomenon known as the Coloumb Nanoradiator effect [80]. To test whether this secondary radiation could disrupt Aβ aggregates, magnetite-bound Aβ fibrils were attached to the bottom of a well, and cortical neuron cells were cultured on top of this matrix. Irradiating these cultures with a 100 MeV proton beam using a total dose of 2 or 4 Gy induced a significant decrease in the density of the fibrils, which was accompanied by a decrease in fibril-bound magnetite, while neurons did not suffer any vital damage.

### 4.2. Animal Models

In a mouse AD model, the effects of ionizing radiation in Aβ plaques were analyzed [89]. The right brain hemisphere of AD mice was irradiated with X-rays in single-dose or fractionated schemes, with a total dose range of 5–20 Gy. There was a significant decrease in both number and size of Aβ aggregates, being fractionated schemes the most effective: the 2 Gy × 10 fractions plan reached a 78% reduction in the number of plaques. Administering the dose in small fractions induced fewer negative effects in normal brain tissue. Although they detected signs of neuroinflammation 24–48 h post-irradiation (e.g., microglia activation), RT improved cognition of AD mice. A dose of 2 Gy per day for 5 consecutive days (total dose of 10 Gy) in the same AD model reduced by 20% the number of Tau tangles, in strong correlation with the decrease in Aβ plaques [90]. However, there are no data about longevity and other disease-related outcomes in this model beyond 8 weeks after irradiation. The protective effect of RT in AD models was partially confirmed later [75]. A dose of 9 Gy administered in 5 fractions of 1.8 Gy had no significant effect on the burden of Aβ plaques in Aβ-overexpressing transgenic mice 4 days after irradiation. However, a significant reduction was verified in synaptic loss, neurodegeneration and neuroinflammation (i.e., microglia and astroglia activation). The same group reported a significant decrease in Aβ accumulation 8 weeks after administering 10 Gy in 2 Gy fractions to 5XFAD transgenic mice carrying five mutations associated with familiar AD [76]. A single 1.76 Gy total-brain irradiation with a Co-60 source in a swine model of AD produced significantly lower levels of hyperphosphorylated Tau (typical of Tau tangles in AD) in the frontal cortex and hippocampus, and of amyloid precursor protein and GAP43 (a marker associated with neosynaptogenesis, neuroplasticity and axonal regeneration) in the cerebellum [91]. There was no evidence of microglia activation, necrotic–ischemic vascular lesions, myelin loss or nuclear damage in neurons or glial cells. However, there was an increase in the number of astroglia in the hippocampus. Low-dose γ irradiation in an Aβ_1-42_-expressing Drosophila model of AD suppressed their morphological defects, motor dysfunction and cell death, but did not alter the survival rates and longevity [92]. Unfortunately, Aβ_1-42_ aggregation was not evaluated in this model.

In a PD mouse model, irradiation with a *γ*-ray source promoted neuroprotective effects [93]. Mice were pre-irradiated with a total dose of 1.5 Gy given in 0.25 Gy fractions once a week before Parkinsonism was induced with reserpine. Radiation diminished reserpine-induced oxidative stress and iron levels (pro-oxidant), increased glutathione levels and quinone oxidoreductase activity (antioxidant) and ameliorated mitochondrial dysfunction. In a retinitis pigmentosa mouse model, photoreceptor cell apoptosis decreased when the total dose was less than 2 Gy, but a dose of 0.64 Gy with a 0.025 Gy/min rate was the most effective plan [45]. This outcome was associated with a striking up-regulation of the *Peroxiredoxin 2* antioxidant gene (563% for the 0.025 Gy/min, 0.64 Gy plan), not verified in high-dose RT treatments. Moreover, multiple low-dose RT sessions enhanced the effects against secondary degeneration in the inner nuclear layer of the retina and in the cone photoreceptor cells.

Charged particle irradiation has been conducted in AD mice models to study the effect of space radiation in astronauts [94,95]. *APP/Presenilin 1* (*PS1*) transgenic mice were irradiated with 0.1, 0.5 and 1 Gy using 150 MeV protons in the Bragg Peak plateau region [94]. A significant decrease in Aβ plaques was verified in the dorsal cortex 9 months post-irradiation with 0.5 Gy, while a higher dose (1 Gy) induced an increase. AD mice (3 × Tg) were also irradiated with highly charged ions (^56^Fe, ^28^Si, and solar particle events) and amyloid and tau pathology was analyzed 7 months later [93]. Amyloid load and the neuroinflammatory gene expression signature were significantly reduced after 2 Gy irradiation with solar particle events in the subiculum of female mice. A trend towards less Tau phosphorylation was verified in female mice, but there were no significant differences nor were there cognitive changes. Whole-body irradiation of AD-transgenic mice using 0.1 or 0.5 Gy ^56^Fe ions induced a reduction in insoluble Aβ_1-40_ levels, β-sheet conformation and microglial activation in females two months later. However, there were no significant changes in locomotor activity, motor learning, grip strength and cognition.

To target endogenous brain magnetite, with the purpose described above, the whole brain of AD mice was irradiated with a single dose of 2 Gy or a fractionated scheme of 4 Gy (2 + 2 Gy with one month of interval) using a 100 MeV proton beam [87]. The number of plaques and ferrous iron foci in the cortex and hippocampus was reduced by 72% and 87%, respectively, 7 days after irradiation, and the area they occupied decreased by 66% and 85%, respectively. Degradation of the peptide matrix with a conformational change of the β-sheets occurred only in the fibrils associated with magnetite, implying an increase in radiation sensitivity when amyloids are aggregated with the natural nanoparticle. Most importantly, cognitive functions improved significantly with no significant damage induced in the normal brain tissue.

### 4.3. Clinical Trials

Probably the first indication that ionizing radiation could be beneficial for AD and PD in humans was published in 2016 as a case report, and in the follow-up articles by the same group until 2021 [96,97,98,99]. Briefly, an 81-year-old AD patient with very advanced disease received five computed tomography (CT) scans over three months (0.04 Gy each), and her cognitive and physical status improved significantly. Since her husband had PD, they decided to carry out a pilot study with him too administering six CT scans each with a dose of 0.04 Gy, with clear qualitative improvements in tremor and constipation soon after each scan [97]. Tremor disappeared to the point that he did not need to use medication to control it for a period of time. Other symptoms, such as hearing loss or Fuch’s endothelial cornea dystrophy including corneal edema, also improved after scans [98]. Follow-up evaluations until 2017 indicated some biphasic responses to radiation, where the patient alternated seasons with improvement and seasons with regression in her symptoms. Based on these promising results, a pilot clinical trial was approved, and enrolled four AD patients [99]. Every 2 weeks, these patients received a CT scan (0.08, 0.04 and 0.04 Gy), and three of them showed immediate recovery in qualitative terms, especially after the 0.08 Gy CT. Summarized information about these patients’ treatment can be found in Table 2. Presently, there are scant quantitative data about the evolution of these patients. It would be interesting to analyze more objective parameters or markers related to neuroinflammation, oxidative stress and amyloidogenesis. Some markers can be detected in blood, and brain response could be evaluated by magnetic resonance imaging.

Currently, clinical trials are going on at the University Hospital of Geneva (Swizterland), the William Beaumont Hospitals in Michigan (USA), and the Kyung Hee University Hospital at Gangdong (South Korea). The first study is at the recruitment phase and intends to enroll 20 patients diagnosed with mild to moderate AD [100]. The subjects will be divided in two experimental arms: 10 patients will undergo observation only, and the other 10 patients will undergo low-dose RT with 5 fractions of 2 Gy for a total dose of 10 Gy. The subjects will be evaluated with positron emission tomography (PET) scans (before irradiation and 8–12 weeks after RT) and neurocognitive tests (before irradiation and 6 months after treatment). Possible side effects will be also evaluated 12 months after RT. The William Beaumont Hospitals study intends to recruit 30 patients diagnosed with AD and to administer 5 daily fractions of 2 Gy to half of the patients and 10 daily fractions of 2 Gy to the other half [101]. Patients will be evaluated in terms of treatment toxicity and neurocognitive function, and perform PET scans 1.5, 3, 6 and 12 months after irradiation. Unfortunately, this study is suspended for now due to staff and budget limitations. The South Korean study is at the recruitment stage and intends to enroll 10 patients diagnosed with mild or moderate AD [102]. The study was designed to administer a fractionated whole brain radiation dose of 1.8 Gy in 3 or 5 fractions, with a total dose of 5.4 or 9 Gy (5 patients for each treatment condition). Subjects will undergo neurocognitive testing and PET scanning 6 months post-RT.

A similar clinical trial started at Virginia Commonwealth University (USA), but it was first suspended and eventually terminated due to the COVID-19 pandemic [103]. Some results were published regarding the only five subjects enrolled in the study. These subjects underwent a RT course of 10 Gy in 5 daily fractions of 2 Gy and were evaluated before the treatment and 12 months later. The report shows very small differences regarding neurocognitive and psychological functions, and improvement in quality of life. PET scans to evaluate amyloid plaque size, number and location were not provided. To the best of our knowledge, no studies on the effect of RT on other known neurodegenerative disorders such as PD or HD have been reported to date.

## 5. Conclusions

Protein misfolding and aggregation in amyloid structures, oxidative stress and neuroinflammation are common hallmarks of neurodegenerative disorders, including AD, PD, HD and prion disorders. Their role is still a matter of intense debate, especially after the failure of antioxidants and anti-amyloidogenic drugs in clinical trials. However, existing evidence indicates that certain RT strategies could diminish or prevent these hallmarks as well as improve behavioral/clinical symptoms. RT showed successful results in the treatment of extra-cranial amyloidosis, similar in many molecular and histopathological aspects to AD, PD and other neurodegenerative disorders. Conventional high-dose RT is most likely too aggressive for these disorders, where the goal is to prevent neuronal death, not to kill cells. However, low-dose RT has shown promising results and new modalities, such as FLASH or proton therapy, alone or in combination with radiation enhancers such as magnetite, could become promising treatments for neurodegenerative disorders involving amyloidogenesis. Interestingly, in vivo studies performed with FLASH irradiation of the brain in mice demonstrated a reduction in cognitive deficits due to less activation of microglial inflammation and a relative preservation of neurogenesis in comparison with conventional RT [104,105,106]. Further research is needed to understand the cellular and molecular mechanisms underlying the beneficial effects of ionizing radiation in diseases associated with the central nervous system.

## Figures and Tables

**Table 1 ijms-23-12265-t001:** List of reviewed studies describing the use of RT with fractionated dose schemes in amyloidosis treatment.

Amyloidosis	Irradiation Conditions	Nr. of Patients	Follow-Up Time	Results	Reference
Orbital	30–34 Gy/15–17 fractions/6 MV photons	2	2 and 6 years	Improvement of proptosis and eye movements and no progression of the disease	[14]
Eyelid	20–30 Gy/10–20 fractions/6 MV photons	4	1–2 years	No disease progression of the disease 1 year after treatment, but some increase in amyloid deposition 1 year later	[15]
Nasopharyngeal	70 Gy/25 fractions/6 MV photons (intensity modulated RT)	1	1 year	Mass decreased 3 months after treatment and disappeared 1 year later	[64]
Laryngeal	45 Gy/25 fractions/energy not defined	1	11 months	Voice was back to normal, and the mass disrupting vocal’s function disappeared	[59]
Tracheobronchial	20 Gy/10 fractions in 2 weeks/4 MV photons Repeated scheme after 6 months	1	1.5 years	The irradiated areas were almost normal in appearance. The patient was free of symptoms	[13]
Tracheobronchial	20 Gy/10 fractions/6 and 10 MV photons	1	21 months	Improvement 6 months later, and bronchoscopy revealed a reduction in amyloid deposits 11 months after therapy	[62]
Tracheobronchial	24 Gy/12 fractions daily/6 MV photons	1	1.5 years	Aeration improved significantly and the mucosa of the trachea was almost restored to normal. However, there was still some thickening in the lower lobes and bronchus intermedius	[61]
Tracheobronchial	24 Gy/12 fractions/6 MV photons	1	9 months	Bronchoscopy and chest X-ray revealed no disease progression, and the patient symptoms were improved	[63]
Pulmonary	24 Gy/12 fractions over 18 days/6 MV photons	3	3.5–4.5 years	Pulmonary tests and radiological images showed improvements, which were accompanied with fewer symptoms	[65]
Urinary Bladder	24 Gy/12 fractions/6 and 18 MV photons	1	7 months	Bladder was normal by cystoscopy and without signs of amyloidosis	[66]

**Table 2 ijms-23-12265-t002:** Summary of the treatment details for the patients that underwent CT scans.

Patient	Disease	Age (years)	Nr. of Scans	Total Administered Dose (Gy)
1	AD	81	11	0.447
2	PD	n.a.	6	0.240
3	AD	88	4	0.165
4	AD	90	4	0.175
5	AD	84	4	0.162
6	AD	82	4	0.161

## Data Availability

Not applicable.

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
