# Peer review of "Radiation as a Tool against Neurodegeneration—A Potential Treatment for Amyloidosis in the Central Nervous System"

_ijms, 2022, doi:10.3390/ijms232012265_

Round 1
Reviewer 1 Report
The authors in this article have very-well tried to study and document the effect of Radiotherapy on neurodegenerative disorders such as AD, PD or HD which has not been reported earlier. The article is well-written and included extensive clinical surveys which suggests that patients who undergo computed tomography (CT) scans for other non-abrasive diseases i.e. non-cancer or non-lymph, are unknowingly getting treated for dementia or cancer related disorders. It would interesting for the authors and readers to supplement there work by comparing the Energy of ionizing radiation used in different scenarios of diseases and patient age.
The authors could tabulate ionizing radiation data in the form of 1) total radiation dose exposed, 2) absorbed radiation dose, 3) effective radiation dose during CT exposure.
Author Response
Firstly, we would like to thank the reviewer for the time spent and for the careful review of our article. We are grateful for the insightful comments on and valuable improvements to our paper. We have carefully considered the comments and tried our best to address them. We hope the manuscript after careful revisions meet your high standards. Please see below the detailed response to your comments and concerns. All page numbers refer to the revised manuscript file with tracked changes.
Point 1: The article is well-written and included extensive clinical surveys which suggests that patients who undergo computed tomography (CT) scans for other non-abrasive diseases i.e. non-cancer or non-lymph, are unknowingly getting treated for dementia or cancer related disorders. It would be interesting for the authors and readers to supplement there work by comparing the Energy of ionizing radiation used in different scenarios of diseases and patient age. The authors could tabulate ionizing radiation data in the form of 1) total radiation dose exposed, 2) absorbed radiation dose, 3) effective radiation dose during CT exposure.
Response 1: In terms of the radiation dose administered with the CT scans the information available in the references is already mentioned in the article, with exception to the dose administered in the case of the Parkinson Disease patient and for that reason we added at line 285 after “a pilot study with him too” the following: “administering six CT scans each with a dose of 0.04 Gy”. Trying to correspond to your suggestion we also added table 2 that summarizes the available information about the CT trials and we presented the table in line 295 with the sentence “Summarized information about these patients’ treatment can be found in table 2”. Unfortunately, the references have no information about the absorbed radiation dose and the effective radiation dose and for that reason it is not possible to added it to the table. In addition, we check the total radiation dose that the patients underwent is mentioned in table 1 for the extracranial amyloidosis cases. To have the information about the total radiation dose administered in all the cases mentioned we added at line 192, before the reference [76], the following: “with a total dose of 0.5, 1, 2, 3.8 or 5.25 Gy”. At line 211 we added “using a total dose of 2 or 4 Gy” before the expression “induced”. We also added at line 223 “(total dose of 10 Gy)” before “in the same AD model”.
Reviewer 2 Report
This is a very well written review of the current status of studies utilizing low-dose radiation to treat neurodegenerative diseases of the brain. This is a relatively new area of research that is gaining interest despite the long held view that irradiating the brain can be nothing other than damaging.
I have no real comments except for the final conclusions. I doubt whether FLASH would ever be applicable to this scenario as it is only effective in relatively high doses and because of its mechanism of action it may not be effective in these low-dose studies. Similarly, proton beams could be used but technically there is no need to go to such an expensive option when photon beams are adequate to deliver relatively simple whole-rain irradiation.
Author Response
Firstly, we would like to thank the reviewer for the time spent and for the careful review of our article. We are grateful for the insightful comments on and valuable improvements to our paper. We have carefully considered the comments and tried our best to address them. We hope the manuscript after careful revisions meet your high standards. Please see below the detailed response to your comments and concerns. All page numbers refer to the revised manuscript file with tracked changes.
Point 1: I doubt whether FLASH would ever be applicable to this scenario as it is only effective in relatively high doses and because of its mechanism of action it may not be effective in these low-dose studies. Similarly, proton beams could be used but technically there is no need to go to such an expensive option when photon beams are adequate to deliver relatively simple whole-brain irradiation.
Response 1: In order to elucidate better the readers about these new RT modalities we added at line 343, before the final sentence, the following: “Interestingly, in vivo studies performed with FLASH irradiation of the brain in mice demonstrated a reduction of cognitive deficits due to less activation of microglial inflammation and a relative preservation of neurogenesis in comparison with conventional RT [104-106].” Regarding the use of proton beams we understand the reviewer point of view and although we exposed in the article that proton therapy can promote the reduction of the amyloid plaques probably, we believe that further research is needed to understand the real value of this technique for neurodegenerative disorders.
In concordance with the changes made we have included the following 3 references:
- Montay-Gruel, P.; Petersson, K.; Jaccard, M.; Boivin, G.; Germond, J. F.; Petit, B.; Doenlen, R.; Favaudon, V.; Bochud, F.; Bailat, C.; et al. Irradiation in a flash: Unique sparing of memory in mice after whole brain irradiation with dose rates above 100Gy/s. Radiother Oncol 2017, 124, 365-369.
- Montay-Gruel, P.; Acharya, M. M.; Gonçalves, J. P.; Petit, B.; Petridis, I. G.; Fuchs, P.; Leavitt, R.; Petersson, K.; Gondré, M.; Ollivier, J.; et al. Hypofractionated FLASH-RT as an Effective Treatment against Glioblastoma that Reduces Neurocognitive Side Effects in Mice. Clin Cancer Res 2021,27, 775-784.
- Simmons, D. A.; Lartey, F. M.; Schüler, E.; Rafat, M.; King, G.; Kim, A.; Ko, R.; Semaan, S.; Gonzalez, S.; Jenkins, M.; et al. Reduced cognitive deficits after FLASH irradiation of whole mouse brain are associated with less hippocampal dendritic spine loss and neuroinflammation. Radiother Oncol 2019, 139, 4-10.